# The HD-ZIP Gene Family in Watermelon: Genome-Wide Identification and Expression Analysis under Abiotic Stresses

**DOI:** 10.3390/genes13122242

**Published:** 2022-11-29

**Authors:** Xing Yan, Zhen Yue, Xiaona Pan, Fengfei Si, Jiayue Li, Xiaoyao Chen, Xin Li, Feishi Luan, Jianqiang Yang, Xian Zhang, Chunhua Wei

**Affiliations:** 1State Key Laboratory of Crop Stress Biology in Arid Areas, College of Horticulture, Northwest A&F University, Yangling, Xianyang 712100, China; 2College of Horticulture and Landscape Architecture, Northeast Agricultural University, Harbin 150030, China; 3State Key Laboratory of Vegetable Germplasm Innovation, Tianjin 300384, China

**Keywords:** HD-ZIP transcription factors, phylogenetic evolution, expression pattern, abiotic stresses

## Abstract

Homeodomain-leucine zipper (HD-ZIP) transcription factors are one of the plant-specific gene families involved in plant growth and response to adverse environmental conditions. However, little information is available on the HD-ZIP gene family in watermelon. In this study, forty *ClHDZs* were systemically identified in the watermelon genome, which were subsequently divided into four distinctive subfamilies (I–IV) based on the phylogenetic topology. HD-ZIP members in the same subfamily generally shared similar gene structures and conserved motifs. Syntenic analyses revealed that segmental duplications mainly contributed to the expansion of the watermelon HD-ZIP family, especially in subfamilies I and IV. HD-ZIP III was considered the most conserved subfamily during the evolutionary history. Moreover, expression profiling together with stress-related *cis*-elements in the promoter region unfolded the divergent transcriptional accumulation patterns under abiotic stresses. The majority (13/23) of *ClHDZs* in subfamilies I and II were downregulated under the drought condition, e.g., *ClHDZ4*, *ClHDZ13*, *ClHDZ18*, *ClHDZ19*, *ClHDZ20*, and *ClHDZ35*. On the contrary, most HD-ZIP genes were induced by cold and salt stimuli with few exceptions, such as *ClHDZ3* and *ClHDZ23* under cold stress and *ClHDZ14* and *ClHDZ15* under the salt condition. Notably, the gene *ClHDZ14* was predominantly downregulated by three stresses whereas *ClHDZ1* was upregulated, suggesting their possible core roles in response to these abiotic stimuli. Collectively, our findings provide promising candidates for the further genetic improvement of abiotic stress tolerance in watermelon.

## 1. Introduction

HD-ZIP transcription factors are a type of protein containing the homeodomain (HD)-leucine zipper (LZ) motif, which are prevalent in the plant kingdom and play important roles in growth and developmental processes. According to the structural characteristics, sequence conservations, and functional properties, the HD-ZIP gene family can be further divided into four subfamilies, i.e., HD-ZIP I, HD-ZIP II, HD-ZIP III, and HD-ZIP IV [1,2]. In addition to the HD and LZ domains, HD-ZIP II proteins contain an extra conserved CPSCE motif (Cys, Pro, Ser, Cys, and Glu) downstream of the LZ domain compared with subfamily I [1,2]. Although proteins from subfamilies III and IV are characterized by an additional START (steroidogenic acute regulatory protein-related lipid transfer) domain and a conserved SAD (START-associated domain along with HD and LZ domains), HD-ZIP III can be distinguished from HD-ZIP IV by the presence of C-terminal MEKHLA domain [2,3,4]. Previously, a genome-wide survey of the HD-ZIP gene family has been performed in various species, such as 48 members of Arabidopsis [1], 40 of cucumber [5], 49 of tomato [3], 45 of sesame [4], 57 of cassava [6], and 48 of rice [7].

Numerous evidence has confirmed that HD-ZIP TFs participate in a wide variety of developmental processes [1,2]. To date, HD-ZIP I genes are generally known for participating in plant growth and development, as well as in responses and tolerance to abiotic stresses, including drought, salinity, and cold stimuli [1,2,4,5,8]. For example, the paralogous genes *AtHB7* and *AtHB12* in Arabidopsis can repress the expression of two ABA receptors, PYL5 and PYL8, thereby functioning as mediators of the negative feedback effect in ABA signaling in response to water deficit [8]. In the case of the subfamily HD-ZIP II, the members are renowned for their functions in shade avoidance, leaf polarity, and abiotic stresses, including light, drought and salt stimuli [1,2,4]. For instance, the genes *AtHB2* and *AtHB4* act together to control early exit from proliferation during leaf development under canopy shade [4,9]. HD-ZIP III proteins have been reported to mainly function as a developmental regulator of apical meristem, vascular system formation, and normal organ polarity [1,2,4,5]. In Arabidopsis, for example, the overexpression of the auxin-regulated gene *AtHB8* can promote vascular cell differentiation, while the decreased transcriptional level of the miR166-mediated gene *AtHB15* can alter the vascular system with expanded xylem tissues and inter-fascicular regions [10,11]. In addition, the subfamily HD-ZIP IV genes generally participate in anthocyanin accumulation, epidermal cell differentiation, trichome formation, and root development [1,2,6], e.g., *ANL2* affecting anthocyanin accumulation in subepidermal cells and root development [12] and *atml1 pdf2* double mutants exhibiting severe defects in shoot epidermal cell differentiation [13].

Watermelon (*Citrullus lanatus* L.) is cultivated worldwide because of its important economic and horticultural performance. However, the production and quality of watermelon is threatened by numerous environmental stresses, including heavy metal toxicity, drought, salinity, cold, and pathogen and/or virus invasion. To adapt to and resist these adverse conditions, plants have evolved a series of defense mechanisms. Transcription factors (TFs) have been well-known to play vital roles in plant responses to biotic and abiotic stimuli. Furthermore, with the availability of a high-quality watermelon reference genome [14,15], a series of TFs have been genome-widely annotated, such as MYB [16] and WRKY [17]; however, the detailed identification of HD-ZIP gene family as well as the response to abiotic stresses is still unclear for watermelon. In this study, we performed a genome-wide identification of HD-ZIP genes in watermelon and systemically analyzed their gene structures, conserved motifs, *cis*-elements, and phylogenetic relationship. Moreover, the expression patterns of members in subfamilies I and II under abiotic stresses were also investigated. Our results provide new insights for further functional studies of stress-responsive HD-ZIPs in watermelon.

## 2. Materials and Methods

### 2.1. Identification and Biochemical Characterization of HD-ZIP Genes in Watermelon

For the genome-wide identification of HD-ZIP genes in watermelon, the protein sequence of reference genome 97103 (V2) were downloaded from Cucurbit Genomics Database (CuGenDB, http://cucurbitgenomics.org/, accessed on 29 December 2021). Additionally, the related protein sequences of the HD-ZIP genes were retrieved for Arabidopsis (48 *AtHDZs*) and cucumber (40 *CsHDZs*) according to recently published research [1,5]. Using the BlastP program, the *AtHDZs* and *CsHDZs* were used as queries to identify HD-ZIP homologs in watermelon. The reliability of the candidate genes was verified through searching against the Pfam and SMART databases, and that with both the conserved HD and LZ domains were retained for further analyses.

The theoretical molecular weights (MW), isoelectric points (pI), and grand average of hydropathicity (GRAVY) of *ClHDZs* were examined using the ProtParam tool of the Expasy online server (https://web.expasy.org/protparam/, accessed on 14 May 2021). The genomic distribution of *ClHDZs* on the chromosome was drawn with the software TBtools (v 1.100) [18].

### 2.2. Phylogenetic and Syntenic Analyses of ClHDZs

Full-length protein sequences of the HD-ZIP genes from watermelon, cucumber, and Arabidopsis were aligned using the software Muscle under default parameters [19], which were subsequently employed to construct phylogenetic trees via MEGA7.0 using the neighbor-joining method (1000 bootstrap replicates) [20]. The destination tabular (-m 8) files generated by BlastP program, together with the corresponding GFF profiles, were served as the input documents for the MCScanX software to analyze the synteny relationships [20]. The collinear results were visualized using CIRCOS software (http://circos.ca/, accessed on 25 October 2021).

### 2.3. Prediction of Gene Structure, Conserved Motifs, and cis-Regulatory Elements

The exon-intron organization of *ClHDZs* was displayed with the online tool GSDS2.0 (http://gsds.gao-lab.org/ accessed on 25 October 2021) based on their corresponding genomic and cDNA sequences. The conserved motifs in *ClHDZs* were predicted by the online program MEME (http://meme-suite.org/tools/meme accessed on 25 October 2021), setting the maximum number of motifs to 20 and motif width of 6–50. To obtain the *cis*-regulatory elements of *ClHDZs*, approximately 2.0 Kb of sequence upstream of the ATG start codon was retrieved and submitted to the PlantCARE server for scanning.

### 2.4. Plant Materials and Treatments

The watermelon inbred line ‘YL’ was used for the materials, which was provided by the Cucurbits Germplasm Resource Research Group at Northwest A&F University, Yangling, Shaanxi, China. The germinated seeds were sown in plastic pots filled with commercial peat-based compost (one seed for each pot) and cultivated under natural conditions in a greenhouse at Northwest A&F University. Finally, seedlings at the four-leaf stage were subjected to treatment under drought, low-temperature, and salt stimuli.

To simulate a natural drought treatment [17,20], the plants were not irrigated for 8 days, and leaves were collected 0, 2, 4, 6, and 8 days post-treatment (dpt). For the low-temperature treatment, plants were kept in a growth chamber at 4 °C, and leaves were sampled 0, 6, 12, 24, and 48 h post-treatment (hpt). For the salinity treatment, seedlings were irrigated with 400 mN of NaCl solution (100 mL per plant), followed by harvesting leaves 0, 8, 18, 30, 42, and 54 h post-treatment (hpt). For each treatment, samples collected at 0 dpt or hpt were used as the control. At each time point, the leaves from three different plants were pooled together, which were immediately frozen in liquid nitrogen and stored at −80 °C until further analysis. In this study, three biological replicates were used in all treatments.

### 2.5. RNA Extraction and Quantitative Real-Time RT-PCR

Following the manufacturer’s instructions, the total RNA was extracted from the samples using the RNASimple Total RNA Kit (TIANGEN, Beijing, China) and the cDNA was synthesized using the FastKing RT Kit with gDNase (TIANGEN, China).

The qRT-PCR amplification was performed on a StepOnePlus Real-Time PCR system (Applied Biosystems, Foster, MN, USA). Using the housekeeping gene *ClACT* (*Cla007792*) as the internal reference, the relative expression level for each selected gene was calculated using the 2^−∆∆Ct^ method, as described in our previous study [21,22]. All primers of *ClHDZs* are listed in Appendix A. Furthermore, the expression level was log2-transformed to obtain heatmaps via the software Tbtools (v 1.100) [18].

## 3. Results

### 3.1. Genome-Wide Identification of HD-ZIP Genes in Watermelon

After discarding redundant sequences, a total of 40 HD-ZIP genes were genome-widely identified in watermelon. To clarify, these HD-ZIP genes were renamed from *ClHDZ1* to *ClHDZ40* based on their chromosome locations (Table 1). Moreover, other features of the *ClHDZs* are also provided in Table 1, such as the protein length, molecular weight (MW), grand average of hydropathicity (GRAVY), and isoelectric points (pI). The protein length varied from 181 to 872 aa, whereas the isoelectric point values ranged from 4.62 to 9.65. All identified *ClHDZs* were naturally hydrophilic, with the grand average of hydropathicity (GRAVY) values being below zero (Table 1). As expected, the *ClHDZs* were unevenly distributed on chromosomes (Figure 1), such as chromosome 01 harboring eight members but none being harbored on chromosome 08.

### 3.2. Evolutionary and Syntenic Analyses of HD-ZIP Gene Family

To explore the evolutionary relationship of the HD-ZIP gene family, a phylogenetic tree was constructed using the 40 *ClHDZs* (Figure 2A). Supported by high bootstrap values, four distinct groups were divided and named as subfamilies I–IV. Among them, subfamily I was the largest, containing 14 *ClHDZs*, followed by subfamilies II and IV, which had 11 and 9 members, respectively. Subfamily III was the smallest subfamily, only harboring six HD-ZIP homologs. Tandem and segmental duplications have been reported to play essential roles in the expansion and function of a gene family [20,23]. To reveal the diverse expansion of *ClHDZs*, duplication events were subsequently analyzed using the software MCScanX. As a result, a total of 14 pairs of segmental duplication events were detected (Figure 3A and Appendix A). Notably, approximately 19 *ClHDZs* from subfamilies I and IV were situated in the collinear regions; it is inferred that the segmental duplications mainly contributed to the expansion of watermelon HD-ZIP family, especially in subfamilies I and IV.

For the comparative evolutionary analysis of this gene family in plants, a phylogenetic tree was constructed using the protein sequences of the 40 *ClHDZs*, as well as that from cucumber (40 *CsHDZs*), Arabidopsis (48 *AtHDZs*), and rice (48 *OsHDZs*) (Appendix A). Without a doubt, four distinct subfamilies were observed with homologs from all the four species, suggesting conserved basal architectures in the evolutionary process. Moreover, syntenic analyses revealed that approximately 52 collinear regions were found between watermelon and cucumber, with the largest one spanning about 9.15 Mb and containing five HD-ZIP homologs (Figure 3A and Appendix A). Notably, 35 *ClHDZs* and 36 *CsHDZs* were found to be located in the corresponding segmental duplication regions in watermelon and cucumber, respectively. Meanwhile, a total of 57 and 41 collinearity paralogs were found between Arabidopsis and watermelon and cucumber, respectively, involving more than 67.0% of the HD-ZIP homologs in each genome. Compared with the average segmental size between watermelon (~2.02 Mb) and cucumber (~1.36 Mb), the size between Arabidopsis and two cucurbit crops were much smaller, i.e., the average size for Arabidopsis (~0.25 Mb), watermelon (~1.01 Mb), and cucumber (~0.52 Mb) (Appendix A).

### 3.3. Gene Structure and Conserved Motif of ClHDZs

To obtain further insight into the evolutionary history of *ClHDZs*, gene structures and intron numbers were comparatively depicted based on their gene annotations and genomic sequences (Figure 2B). In subfamily I, nearly half of the members contained three exons with the intron phase pattern ‘10’, while those in subfamily II generally had four exons, displaying the intron pattern ‘102’. In contrast, all HD-ZIP homologs in subfamily III exhibited a much more complex exon-intron organization, with 18 exons sharing the major intron phase ‘20200000020000200’. As for subfamily IV, six out of the nine members were constituted of 11 exons with the phase pattern ‘0200000100’, while the remaining three homologs contained 10 exons with the intron phase ‘200000100’, possibly due to an exon lost at the 5′ end.

Similar to the exon-intron organization, the distribution of the motifs were also conserved within a particular subfamily but diverse among subfamilies (Figure 2C). Details of the predicted motifs were exhibited in Appendix A. It is worth noting that motifs 1 and 2 corresponding to the HD domain as well as motif 3 corresponding to the LZ domain were discovered in all of the identified *ClHDZs*. In addition, motifs 2, 5, 6, and 8 corresponding to the START domain were shared among *ClHDZs* from subfamilies III and IV, while the MEKHLA domain (motifs 9 and 15) was only discovered in subfamily III. Consistent with the literature [1,2], members in HD-ZIP III and IV possessed many more motifs than members in subfamilies I and II (Figure 2C).

### 3.4. Prediction of cis-Regulatory Elements in ClHDZs

To investigate the potential function of *ClHDZs*, approximately 2.0 kb of genomic sequences upstream of the translation site were retrieved to predict the *cis*-regulatory elements. As shown in Figure 4A, 31 types of *cis*-elements were identified and divided into four categories, including plant growth and development, phytohormones, various stresses, and possible binding sites for transcription factors (TFs). Among them, four types of *cis*-elements were related to plant growth and development, and eight were responsive to phytohormones, such as ABRE (abscisic acid-responsive), CGTCA motif (gibberellin-responsive), ERE (ethylene-responsive), GARE-motif (gibberellin-responsive), and P-box (gibberellin-responsive). The largest category was predicted to be involved in stress responses and consisted of 15 different *cis*-regulatory elements such as ARE (anaerobic-responsive), LTR (low temperature-responsive), WUN motif (wound-responsive element). Moreover, the binding sites for MYB- and MYC-type bHLH TFs were frequently detected in the promoter of *ClHDZs*. Although the number of *cis*-regulatory elements varied dramatically, the stress-related and TF-binding elements occupied the major proportion of all *ClHDZs* (Figure 4B), suggesting their possible functions being involved in stress responses.

### 3.5. Expression Analysis of ClHDZs from Subfamilies I and II under Drought, Cold and Salt Stresses

To date, HD-ZIP I genes are generally known for participating in plant growth and development, as well as in responses and tolerance to numerous abiotic stresses, particularly drought, salinity, and cold stimuli [1,2,4,5,8]. In the case of subfamily HD-ZIP II, members are renowned for their functions in shade avoidance, leaf polarity, and abiotic stresses, including light, drought, and salt stimuli [1,2,4]. Hence, *ClHDZs* in subfamilies I and II were selected to analyze their dynamic response after exposure to drought, cold, and salt stresses using qRT-PCR analysis. Compared with drought stimuli (Figure 5), far more *ClHDZs* could be induced by cold and NaCl treatments, with only a few exceptions being downregulated, such as gene *ClHDZ3* from HD-ZIP I under the cold condition and *ClHDZ14* from HD-ZIP II under the salt stress condition. Under drought conditions, the majority of *ClHDZs* were slightly upregulated two days (2 d) after treatment (Figure 5A), and few of them were subsequently downregulated during the drought duration, e.g., *ClHDZ20* and *ClHDZ30* in subfamily I and *ClHDZ18*, *ClHDZ4*, and *ClHDZ35* in HD-ZIP II. On the contrary, genes *ClHDZ5* and *ClHDZ22* were transiently inhibited at 2 dpt but were significantly enhanced with the advancement of treatment. Moreover, the expression of *ClHDZ23* and *ClHDZ32* showed durable enhancement by drought stimuli. In response to the low-temperature condition (Figure 5B), half of *ClHDZs* exhibited enhanced transcriptional levels, such as *ClHDZ30*, *ClHDZ31, ClHDZ38*, and *ClHDZ39* in HD-ZIP I and *ClHDZ18*, *ClHDZ35*, and *ClHDZ37* in HD-ZIP II. Conversely, only a few members were continuously downregulated, e.g., *ClDHZ3* and *ClHDZ23*. Similarly, most *ClHDZs* were induced by salt stress (Figure 5C), such as *ClHDZ20* and *ClHDZ36* in HD-ZIP I and *ClHDZ1* and *ClHDZ18* in HD-ZIP II. However, only six genes showed obviously downregulated expression during the whole period of treatment, and four of them (*ClHDZ14*, *ClHDZ32*, *ClHDZ4*, and *ClHDZ33*) were gathered together in subfamily HD-ZIP II (Figure 2A). Notably, gene *ClHDZ14* was predominantly downregulated by three stresses while the expression of gene *ClHDZ1* was upregulated by all the treatments, suggesting their possible core roles in watermelon tolerance to these abiotic stresses.

## 4. Discussion

As a plant-specific transcription factor, HD-ZIP gene family members have been widely disseminated in the plant kingdom [24], such as 48 members in Arabidopsis [1], 40 in cucumber [5], 49 in tomato [3], 45 in sesame [4], 57 in cassava [6], 83 in apple [25], and 48 in rice [7]. Here, we identified a total of 40 *ClHDZs* in the watermelon genome (Table 1), which is the same as that observed in cucurbit crop cucumber but relatively lower than that of other species. Similar to the HD-ZIP members in cucumber [5], all identified *ClHDZs* in watermelon were hydrophilic in nature, with the grand average of hydropathicity (GRAVY) values being below zero (Table 1). According to the phylogenetic topology (Figure 2A), the *ClHDZs* were divided into four subfamilies (I, II, III, and IV) in line with the published classifications for other species such as Arabidopsis, cucumber, tomato, and rice [1,3,4,5,6,25]. As the top two largest subfamilies, HD-ZIP I and II contained 14 and 11 members, respectively (Table 1), with exon numbers ranging from 2 to 4 and protein length varying from 181 to 351 aa. In contrast, the HD-ZIP homologs in subfamilies III and IV displayed many more exons and amino acids. Systemic analyses revealed that 21 *ClHDZs* were detected in the segmental region of the watermelon genome, with the majority contributing to the expansion of HD-ZIP subfamilies I and IV (Figure 3A and Appendix A). Notably, as the smallest subfamily, HD-ZIP III contained the least number of genes (6/40), which is consistent with the proportions found in other species, such as cucumber (5/40) [5], Arabidopsis (5/48) [1], and tomato (6/49) [3]. To date, HD-ZIP subfamily III is considered as the most conserved lineage during evolution, members of which function in the plant’s basic growth and development, such as apical meristem and vascular system formation [1]. In this present study, *ClHDZs* in subfamily III also exhibited much more conserved intron-exon organization, intron phase pattern, and motif distribution, which is consistent with observations made with respect to other species [2,3,4,5,25].

HD-ZIP transcription factors have been reported to be involved in responses to a variety of abiotic stress in plants [1,2,4,5], especially members from subfamilies I and II [8,26]. Hence, we focused on the expression patterns of *ClHDZs* from HD-ZIP I and II under drought, low-temperature (4 °C), and salt (Figure 5) stresses. Intriguingly, the majority of *ClHDZs* were induced by low-temperature and salt, whereas some were downregulated under the drought condition, such as gene *ClHDZ18*, *ClHDZ31*, and *ClHDZ35*. Most importantly, some valuable candidate HD-ZIP genes were explored. For instance, gene *ClHDZ14* was predominantly downregulated by the three stresses, but the expression of gene *ClHDZ1* was upregulated by all three treatments. In cucumber, gene *CsHDZ33* from subfamily I showed an induced transcriptional activity under low-temperature, salt, and drought stresses [5]. As its ortholog gene in watermelon, *ClHDF5* also showed an increased expression tendency after exposure to similar adverse conditions (Figure 5). Moreover, the transcriptional levels of gene *ClHDZ3* under drought and salt conditions as well as *ClHDZ18* under cold and salt conditions were obviously accumulated, which was in line with the expression patterns of their orthologs *CsHDZ02* and *CsHDZ22* in cucumber [5]. In Arabidopsis, the paralogous genes *AtHB7* and *AtHB12* were strongly upregulated by drought stress, functioning as mediators of the negative feedback effect on ABA signaling in response to water deficits [8]. According to the phylogenetic analysis, gene *ClHDZ15* was recognized as the closest homolog of *AtHB7* and *AtHB12*, which was also induced at 6 d post-drought treatment (Figure 5A). Similarly, the transcriptional level of *ClHDZ5* was enhanced under drought stress, consistent with the induced expression trend of its homolog *AtHB6* [8,26]. Previously, we have genetic mapped the gene *ClLL1* responsible for lobed leaf in watermelon [27], and the best candidate gene *Cl018360* (HD-ZIP transcription factor) was identified as a ortholog of *AtHB51/LMI1* in Arabidopsis, which is characterized as a vital regulator in leaf proportionality [28]. Here, gene *Cl018360* was also detected and designated as *ClHDZ13* (Table 1). According to the qRT-PCR results, gene *ClHDZ13* was extremely reduced by drought stress (Figure 5A), providing new evidence for this gene’s function in water deficit tolerance. Recently, gene *BnHB1423* (an ortholog of *AtHB51/LMI1*) in *Brassica napus* L. was found to be significantly upregulated by salt stimuli [29], implying its diverse function in plant development and response to adverse environmental conditions.

## 5. Conclusions

In the present study, a total of 40 *ClHDZs* were genome-widely identified in watermelon, which were further divided into four subfamilies. Among them, subfamily III was considered as the most conserved subfamily based on the gene structure and exon-intron organization. Syntenic analyses indicated that the expansion of subfamilies I and IV were possibly due to segmental duplication events. The expression profiling revealed that far more *ClHDZs* could be induced by cold and NaCl treatments compared with drought stimuli. In addition, some valuable candidates were subsequently identified that are possibly involved in responses to abiotic stresses, e.g., drought, low-temperature, and salinity. For example, the expression of gene *ClHDZ23* and *ClHDZ32* showed continuous enhancement with the increase in drought stress, similar to that of gene *ClHDZ35*, *ClHDZ38*, and *ClHDZ39* under the cold condition and *ClHDZ1* and *ClHDZ18* after salt treatment. Moreover, gene *ClHDZ14* was predominantly downregulated by the three stresses, but the expression of gene *ClHDZ1* was upregulated by all treatments. Collectively, these findings may expedite the further functional characterization of *ClHDZs* in response to abiotic stresses.

## Figures and Tables

**Figure 1 genes-13-02242-f001:**
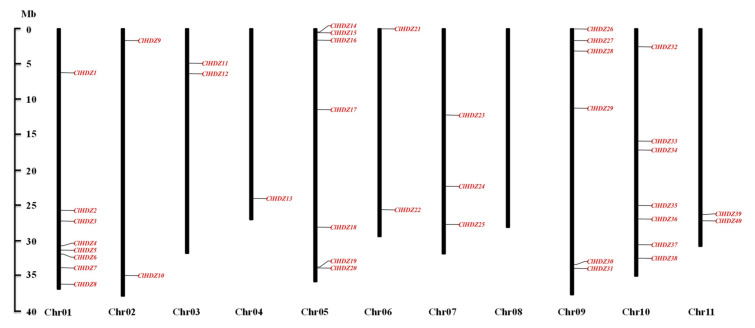
The chromosome distribution of *ClHDZs* in watermelon.

**Figure 2 genes-13-02242-f002:**
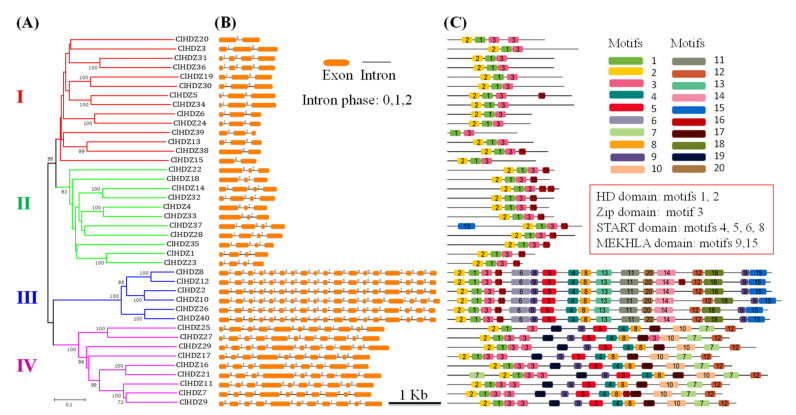
The phylogenetic tree (**A**), gene structure (**B**), and motif distribution (**C**) of *ClHDZs*. All ClHDZs were divided into the four subfamilies I–IV, and the bootstrap values >65 were shown in the phylogenetic tree. Motifs were marked with different colors and accordingly positioned.

**Figure 3 genes-13-02242-f003:**
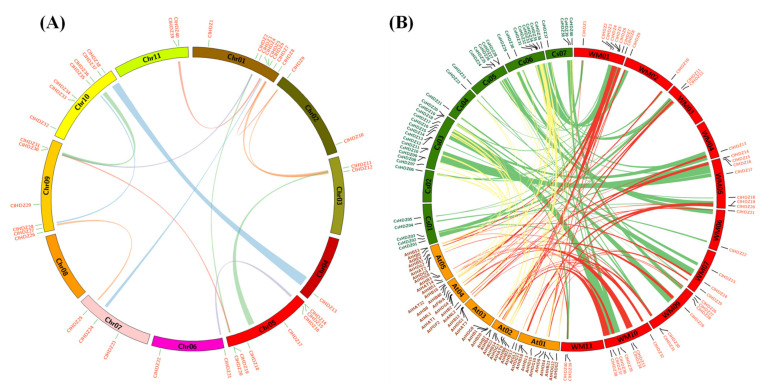
Syntenic analysis of the HD-ZIP genes in watermelon (**A**) as well as among three genomes (**B**), i.e., watermelon (red), cucumber (green), and Arabidopsis (Orange). The approximate position of the HD-ZIP genes on the chromosome were presented by short lines on the circle. Colored curves denote the details of the syntenic region harboring the HD-ZIP genes among the genomes.

**Figure 4 genes-13-02242-f004:**
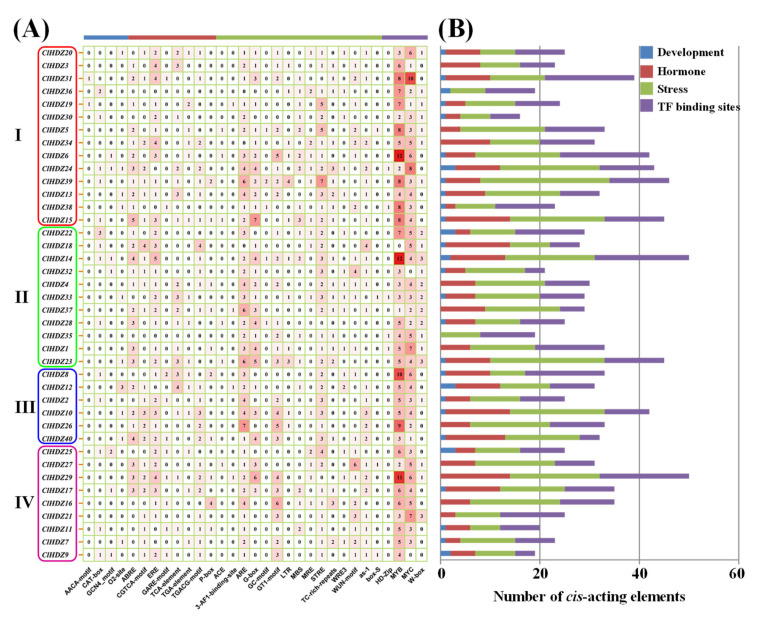
The prediction of cis-elements in promoter regions of *ClHDZs*. The detailed numbers (**A**) and statistical graphs (**B**) are presented, respectively.

**Figure 5 genes-13-02242-f005:**
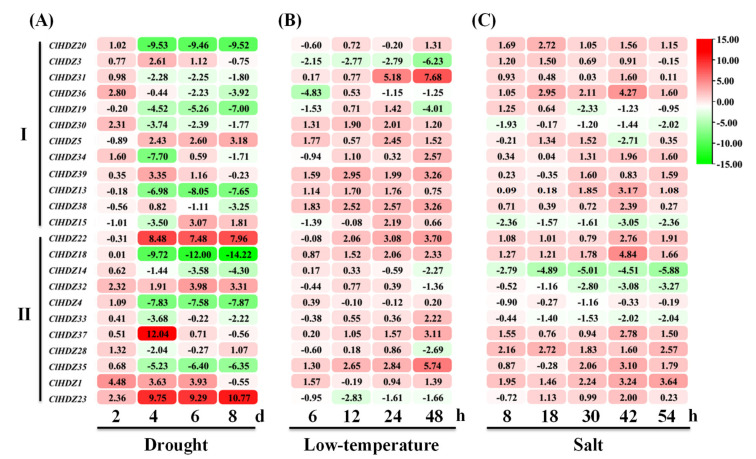
Gene expression heatmap of *ClHDZs* under drought (**A**), low-temperature (**B**), and salt (**C**) conditions. The samples at 0 dpt and 0 hpt were used as the controls, and the numbers present the relative expression levels after log2 transformation.

**Table 1 genes-13-02242-t001:** The gene and protein features of HD-ZIP members in watermelon.

Gene Name	Gene ID	Chromosome Location	Subfamily	Protein Length (aa)	Exon Number	MW(Da)	Gravy	pI
*ClHDZ3*	Cla97C01G013400	Chr01:27240628_27242205	I	341	3	38,075.8	−0.742	4.63
*ClHDZ5*	Cla97C01G017950	Chr01:31344079_31345409	I	324	3	36,813.5	−0.799	4.83
*ClHDZ6*	Cla97C01G018750	Chr01:31916455_31918441	I	220	3	25,182.4	−0.94	8.48
*ClHDZ13*	Cla97C04G076510	Chr04:24039469_24041867	I	224	3	25,795.9	−0.798	6.23
*ClHDZ15*	Cla97C05G080560	Chr05:620815_621585	I	230	2	26,411.5	−0.927	5.82
*ClHDZ19*	Cla97C05G106550	Chr05:33731410_33732703	I	300	3	34,313.1	−1.036	6.38
*ClHDZ20*	Cla97C05G106720	Chr05:33865821_33867128	I	254	2	29,228.5	−0.917	5.09
*ClHDZ24*	Cla97C07G135990	Chr07:22316542_22319234	I	216	3	24,822.1	−0.884	5.86
*ClHDZ30*	Cla97C09G179770	Chr09:33367941_33370759	I	304	3	33,618.7	−0.643	6.46
*ClHDZ31*	Cla97C09G180320	Chr09:33927034_33929071	I	278	4	31,932.2	−0.832	4.62
*ClHDZ34*	Cla97C10G191910	Chr10:17204865_17206214	I	330	3	37,342.1	−0.841	4.81
*ClHDZ36*	Cla97C10G197160	Chr10:26936967_26941299	I	277	4	31,253.6	−0.826	4.9
*ClHDZ38*	Cla97C10G202610	Chr10:32490507_32492602	I	262	2	30,042.6	−0.666	9.65
*ClHDZ39*	Cla97C11G220290	Chr11:26319113_26320758	I	181	3	21,163	−0.882	6.21
*ClHDZ1*	Cla97C01G006340	Chr01:6268951_6272330	II	228	4	25,840.8	−0.593	9.53
*ClHDZ4*	Cla97C01G017140	Chr01:30736614_30737606	II	264	3	29,144.1	−0.655	8.11
*ClHDZ14*	Cla97C05G080370	Chr05:505253_507284	II	291	4	32,306.1	−0.806	7.06
*ClHDZ18*	Cla97C05G098910	Chr05:28098057_28099506	II	267	3	29,247.4	−0.588	8.46
*ClHDZ22*	Cla97C06G123330	Chr06:25635223_25636830	II	279	3	32,054.7	−1	7.67
*ClHDZ23*	Cla97C07G134460	Chr07:12254360_12260018	II	196	4	22,589.9	−0.786	8.68
*ClHDZ28*	Cla97C09G166050	Chr09:3209878_3211125	II	333	4	36,909.1	−0.823	5.69
*ClHDZ32*	Cla97C10G187080	Chr10:2601999_2603584	II	278	4	31,111	−0.801	7.66
*ClHDZ33*	Cla97C10G191810	Chr10:15937459_15938526	II	278	3	30,745.9	−0.676	9.09
*ClHDZ35*	Cla97C10G195530	Chr10:25031280_25034315	II	269	4	30,517.1	−0.933	6.68
*ClHDZ37*	Cla97C10G200510	Chr10:30578990_30581007	II	351	4	38,288.9	−0.574	5.9
*ClHDZ2*	Cla97C01G012720	Chr01:25719610_25727010	III	847	12	92,620.4	−0.079	5.69
*ClHDZ8*	Cla97C01G025160	Chr01:36152833_36158141	III	847	18	92,756.8	−0.142	5.85
*ClHDZ10*	Cla97C02G047220	Chr02:34924600_34930198	III	872	18	95,349.8	−0.067	5.79
*ClHDZ12*	Cla97C03G057570	Chr03:6406462_6413316	III	844	18	92,949.2	−0.11	5.81
*ClHDZ26*	Cla97C09G161990	Chr09:85439_91313	III	837	18	92,308.8	−0.137	6.03
*ClHDZ40*	Cla97C11G221100	Chr11:27192123_27199004	III	842	18	92,544.7	−0.149	5.74
*ClHDZ7*	Cla97C01G021770	Chr01:33838063_33844699	IV	718	10	78,732.3	−0.264	5.76
*ClHDZ9*	Cla97C02G028260	Chr02:1723768_1729493	IV	754	11	82,236	−0.314	5.85
*ClHDZ11*	Cla97C03G056100	Chr03:4929667_4937076	IV	737	10	81,747.9	−0.314	6.03
*ClHDZ16*	Cla97C05G082240	Chr05:1664314_1669186	IV	743	9	81,814.2	−0.277	5.74
*ClHDZ17*	Cla97C05G093030	Chr05:11485675_11489533	IV	710	10	79,039.1	−0.25	6.25
*ClHDZ21*	Cla97C06G109300	Chr06:66818_71658	IV	836	9	89,918.1	−0.246	5.66
*ClHDZ25*	Cla97C07G139950	Chr07:27706301_27714819	IV	771	11	86,037.3	−0.528	6.4
*ClHDZ27*	Cla97C09G164060	Chr09:1721143_1728584	IV	736	11	82,347.4	−0.484	5.99
*ClHDZ29*	Cla97C09G174400	Chr09:11280503_11288355	IV	806	11	89,745.7	−0.391	5.74

## Data Availability

All relevant data can be found within this manuscript and the Appendix A.

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
