# Peer review of "The HD-ZIP Gene Family in Watermelon: Genome-Wide Identification and Expression Analysis under Abiotic Stresses"

_genes, 2022, doi:10.3390/genes13122242_

Round 1
Reviewer 1 Report
stress treatments in the abstract at line number 22-23.
Keywords:
Comment #3 - The keyword ‘phylogeny’ can be included in the keywords.
Introduction:
Comment #4- At line number 53, what is -4? Authors should write full name of gene as AtHB4 instead of writing -4. Similarly, at line number 56, -15 should also be written as AtHB15 instead of -15.
Comment #5- At line number 61 and 71, authors should write word ‘wide’ instead of ‘widely’.
Comment #6- Authors must check the punctuation and comma (,) in the whole text of the manuscript including Introduction section and put it at appropriate region to make the text grammarly sound.
Materials and methods:
Comment #7 - At line number 84, authors should write word ‘candidate genes’ instead of ‘candidates’.
Comment #8- At line number 87, authors should check the abbreviation for molecular weights. They wrote it as WM instead of MW.
Comment #9 - At line number 106, authors should italicise the word cis-’.
Comment #10 – May I ask to authors, what is “national drought treatment”?
Comment #11 – Please remove the word ‘respectively’ at line number 119.
Comment #12 – Please check at line number 157. Is it MScan or MCScanX?
Scientific comments
Comments #13 – Authors must clarify, why the Hidden Markov Model (HMM) profile of the HD-ZIP domain was not generated for identification of HD-ZIP transcription factor gene family in watermelon?
Comment #14- Why authors have not included the functional annotation of HD-ZIP transcription factor gene family in watermelon? So authors can add the same using Blast2go analysis.
Comments #15 – Why do authors have not mentioned the statistical analysis in treatments? It is suggested to incorporate the statistical analysis at p-value at line number 116-124.
Comment #16 – Authors must improve the resolution of Figure 1 and 3. The tags are not visible properly.

Author Response
Response to reviewers:
The study “The HD-ZIP gene family in watermelon: genome-wide identification and expression analysis under abiotic stresses” comprehensively comprised with identification and systematically characterization of HD-ZIP transcription factor gene family in watermelon. Authors have tried all sort of bioinformatics analysis and subsequently wet-lab validation of HD-ZIP transcription factor gene family during drought, low-temperature, and salinity stress in watermelon. The study is well executed with sound experimentations and representations. However, some minor modifications need to be rectified and incorporated by authors to improve the manuscript for global readability and acceptance. The comments observed during review of the manuscripts are as follows:
General comments
Abstract:
Comment #1 - The abstract is well written with sound language. However, ‘40 ClHDZs’ must be written as forty ClHDZs at line number 15.
Many thanks for your kindly advice.
The ‘40 ClHDZs’ had been written as forty ClHDZs in the revised version.
Comment #2 – Authors should mention, how many genes were characterized at expression level during all sort of stress treatments in the abstract at line number 22-23.
Following your advice, the gene number had been added in the abstract of the revised version.
Keywords:
Comment #3 - The keyword ‘phylogeny’ can be included in the keywords.
Yes. The keyword ‘Evolution’ had been rewritten as ‘Phylogenetic evolution’, according to your suggestion.
Introduction:
Comment #4- At line number 53, what is -4? Authors should write full name of gene as AtHB4 instead of writing -4. Similarly, at line number 56, -15 should also be written as AtHB15 instead of -15.
Thank you for your suggestions.
And we had written the full name of genes in the new version of this manuscript.
Comment #5- At line number 61 and 71, authors should write word ‘wide’ instead of ‘widely’.
Yes. We have corrected these mistakes.
Comment #6- Authors must check the punctuation and comma (,) in the whole text of the manuscript including Introduction section and put it at appropriate region to make the text grammarly sound.
Many thanks for your valuable suggestions to improve the quality of this MS.
We had checked the punctuation and comma through the whole manuscript.
Materials and methods:
Comment #7 - At line number 84, authors should write word ‘candidate genes’ instead of ‘candidates’.
Yes. We had corrected this mistake in the new version of this manuscript.
Comment #8- At line number 87, authors should check the abbreviation for molecular weights. They wrote it as WM instead of MW.
Yes. We had corrected this mistake.
Comment #9 - At line number 106, authors should italicise the word cis-’.
Yes. We had corrected this mistake.
Comment #10 – May I ask to authors, what is “national drought treatment”?
Sorry for the wrong description.
It should be ‘natural drought treatment’. And we had corrected this mistake.
Comment #11 – Please remove the word ‘respectively’ at line number 119.
Thanks. We had removed the word ‘respectively’.
Comment #12 – Please check at line number 157. Is it MScan or MCScanX?
It should be ‘MCScanX’. And we had corrected this mistake in the revised version.
Scientific comments
Comments #13 – Authors must clarify, why the Hidden Markov Model (HMM) profile of the HD-ZIP domain was not generated for identification of HD-ZIP transcription factor gene family in watermelon?
Thanks for your comments.
Agree with your points. In numerous researches, the Blastp program and HMMER software were widely used to identify homologous members of gene family in plants. For example, the HD-ZIP genes in cucumber was identified via the HMM profile (Sharif et al., 2020), while that in apple was obtained via Blastp program (Zhang et al., 2022). As we known, watermelon and cucumber are related species in Cucurbitaceae. Hence, we used the published HD-ZIP genes from cucumber and Arabidopsis to identify ClHDZs in watermelon through the Blastp program.
References:
Sharif, R.; Xie, C.; Wang, J.; Cao, Z.; Zhang, H.; Chen, P.; Yuhong, L., Genome wide identification, characterization and expression analysis of HD-ZIP gene family in Cucumis sativus L. under biotic and various abiotic stresses. Int. J. Biol. Macromol. 2020, 158, 502-520
Zhang, Q.; Chen, T.; Wang, X.; Wang, J.; Gu, K.; Yu, J.; Hu, D.; Hao, Y., Genome-wide identification and expression analyses of homeodomain-leucine zipper family genes reveal their involvement in stress response in apple (Malus  ×  domestica). Hortic. Plant J. 2022, 8, (3), 261-278
Comment #14- Why authors have not included the functional annotation of HD-ZIP transcription factor gene family in watermelon? So authors can add the same using Blast2go analysis.
Actually, we performed the GO enrichment analysis with the identified forty ClHDZs. However, we did not find any valuable results. For instance, all the enriched GO terms in Molecular Function are related to the DNA binding (Fig R1), which is recognized the basic function of transcription factors. So, we discarded this part in the original manuscript.
Fig R1. The GO enrichment of ClHDZs
Comments #15 – Why do authors have not mentioned the statistical analysis in treatments? It is suggested to incorporate the statistical analysis at p-value at line number 116-124.
Many thanks for your advice.
Considering the expression levels of ClHDZs were presented as heatmaps, we did not incorporate the statistical analysis at p-value in the MM section.
Comment #16 – Authors must improve the resolution of Figure 1 and 3. The tags are not visible properly.
Yes. We have provided the high resolution Figures in the revised version of this manuscript.
Many thanks for all your valuable suggestions to improve the quality of this MS.

Reviewer 2 Report
The HD-Zip gene family of transcriptional factors is unique to plants and involved in plant growth, development, and in the response to abiotic stress. In Arabidopsis, the HD-Zip gene family is well characterized, whereas in watermelon comprehensive analysis of HD-Zip genes, especially those responsible for combating drought and salinity is lacking. This work by Xing et al adds value; however, I would like to draw attention to the following points
INTRODUCTION
1. In the introduction, elaborate on the role of the HD-Zip gene family in abiotic stress
2. Why only HD-Zip gene subfamilies I and II were included in expression analysis under stress conditions (although recently III subfamilies are found to play role in abiotic stress, reference No 3, Line No 330)
MATERIAL AND METHODS
2.2 Phylogenetic and syntenic analysis of CIHDZs
3. Mention parameter for MUSCLE alignment
2.4 Plant materials and treatments
4. mention the number of plants per pot
5. cite a reference to justify sampling duration for drought, low-temeprature and salt stimuli
RESULTS
3.5 Expression analysis of CIHDZs
6. Elaborate the results of expression analysis under different stresses if possible split section 3.5 into three sub-section one for Drought, one for low-temeprature, and one for salinity
DISCUSSION
7. Discuss the expression pattern of HD-Zip subfamilies I and II with reference to Arabidopsis, rice, etc under stress condition
8. Line 275, Subfamily II instead of III
9. line 285, Drought stress instead of PEG stress
Author Response
Response to reviewers:
The HD-Zip gene family of transcriptional factors is unique to plants and involved in plant growth, development, and in the response to abiotic stress. In Arabidopsis, the HD-Zip gene family is well characterized, whereas in watermelon comprehensive analysis of HD-Zip genes, especially those responsible for combating drought and salinity is lacking. This work by Xing et al adds value; however, I would like to draw attention to the following points
INTRODUCTION
- In the introduction, elaborate on the role of the HD-Zip gene family in abiotic stress
Many thanks for your valuable suggestions.
Following your advice, in the revised version of this MS, we have rewritten the sentences to elaborate the detail functions of HD-ZIP genes under abiotic stresses.
- Why only HD-Zip gene subfamilies I and II were included in expression analysis under stress conditions (although recently III subfamilies are found to play role in abiotic stress, reference No 3, Line No 330)
Thank you for your comments.
According to published researches, the HD-ZIP genes from subfamilies I and II were well known to mainly function in response to various stresses. Hence, we focused on the expression patterns of these ClHDZs (I and II) to obtain some valuable candidate genes for further researches.
MATERIAL AND METHODS
2.2 Phylogenetic and syntenic analysis of CIHDZs
- Mention parameter for MUSCLE alignment
Default parameters were used for Muscle alignment, which had been added in the revised version of this manuscript.
2.4 Plant materials and treatments
- mention the number of plants per pot
One plant for each pot.
And we had added this information in the new version of this manuscript.
- cite a reference to justify sampling duration for drought, low-temeprature and salt stimuli
Yes.
The related reference had been cited in the original manuscript (Line 116 in the original submission, references No 17 and No 20).
RESULTS
3.5 Expression analysis of CIHDZs
- Elaborate the results of expression analysis under different stresses if possible split section 3.5 into three sub-section one for Drought, one for low-temeprature, and one for salinity
Thank you very much for this valuable advice.
According to your suggestions, we had rewritten this part of results in the revised manuscript. However, we did not spilt this conclusion into three sub-sections, because of lacking of enough valuable results for each treatment.
DISCUSSION
- Discuss the expression pattern of HD-Zip subfamilies I and II with reference to Arabidopsis, rice, etc under stress condition
Following your advice, some comparative discussions had been added in the revised version of this manuscript.
- Line 275, Subfamily II instead of III
Many thanks.
We had corrected this mistake in the revised manuscript.
- line 285, Drought stress instead of PEG stress
We had modified this word following your advice.
Many thanks for all your valuable suggestions to improve the quality of this MS.

Reviewer 3 Report
The authors are advised to improve the manuscript in terms of adequate language levels as well as research paper structure.
Many sentences do not have the correct punctuation and it is difficult to read the text.
Figures is not in printable quality. Also some portions of the texts are losing their readability while sizing the image as per text area. Kindly provide better quality figure.
Please check the References in-text and end-list for uniformity in style.
The conclusion you have provided is quite brief and provide sufficient feedback on the main objectives of your study.
Author Response
Response to reviewers:
The authors are advised to improve the manuscript in terms of adequate language levels as well as research paper structure.
Many sentences do not have the correct punctuation and it is difficult to read the text.
Many thanks for you valuable comments. We had carefully revised the language to improve the quality of this manuscript.
Figures is not in printable quality. Also some portions of the texts are losing their readability while sizing the image as per text area. Kindly provide better quality figure.
Thank you for your advice.
We have provided the high resolution Figures in the revised version of this manuscript.
Please check the References in-text and end-list for uniformity in style.
Yes.
We have carefully checked the References of this manuscript, to make sure their uniformity in style.
The conclusion you have provided is quite brief and provide sufficient feedback on the main objectives of your study.
Many thanks.
We have rewritten this part in the revised manuscript.
Thanks again for all your valuable suggestions to improve the quality of this MS.

Reviewer 4 Report
The authors studied 40 ClHDZs where systemically identified genes in watermelon reveal their potential roles in response to 3 diverse abiotic stresses (drought, low-temperature, and salinity). The authors showed great work.
I have found some comments in the manuscript:
-The abstract must be supported with more results
- Figure 4 (A), what is your purpose for the statistics graphs
-The conclusions must be supported with more results
Author Response
Response to reviewers:
The authors studied 40 ClHDZs where systemically identified genes in watermelon reveal their potential roles in response to 3 diverse abiotic stresses (drought, low-temperature, and salinity). The authors showed great work.
Thank you very much for your positive affirmations of our MS.
I have found some comments in the manuscript:
-The abstract must be supported with more results
Thanks for your kindly advice.
The abstract had been modified in the new version of this manscript.
- Figure 4 (A), what is your purpose for the statistics graphs
The Figure 4A is aimed to represent the detail numbers of cis-elements in each ClHDZs.
-The conclusions must be supported with more results
Many thanks.
We have rewritten this part in the revised manuscript.
Thanks again for all your valuable suggestions to improve the quality of this MS.

Round 2
Reviewer 2 Report
All the suggested comments/observations are included in the revised manuscript.
If possible, I request the author to incorporate the method used for sampling (i.e number of biological replicas and technical replicas) for real-time PCR.